# PERFORMANCE VS INTERPRETABILITY TRADE-OFF OF HAND-CRAFTED AND LANGUAGE MODEL FEATURES: THE CASE OF PROTEIN SUPERFAMILY CLASSIFICATION

## ABSTRACT

The newfound rise of protein language models (PLMs) that leverage data and compute has introduced an interesting conflict in computational biology: a trade-off between the high predictive performance of non-interpretable features and the scientific insight that can be gained from interpretable, hand-crafted ones. In this work, we highlight and study this conflict via the task of classifying protein domains into their CATH superfamilies. We train one-vs-all linear SVM classifiers for 45 CATH superfamilies, each characterised by significant class imbalance. We address the class imbalance by using a class-balanced loss function and the arithmetic mean (AM) of specificity and sensitivity for evaluation. Our analysis compares nine feature vector types, which are either non-interpretable embeddings from PLMs or interpretable hand-crafted features. The latter includes amino acid composition (AAC), di- and tri-peptide composition (DPC, TPC), and novel sequence-order (2OAAC, 3OAAC) and structure-based features (OCPC, CSIC). Our results demonstrate that PLM-based features achieve superior test AM scores of 90-99% with low variability, outperforming hand-crafted features by 20-30%. While PLM features yield high classification accuracy, their lack of interpretability obscures the underlying biological determinants. Conversely, the interpretability of hand-crafted features, despite their relatively low performance, can be leveraged to infer sequence and structural characteristics of CATH superfamilies. The proposed hand-crafted CSIC feature stikes a balance between predictive performance and interpretability, because it overfits to a lesser extent. This can be valuable for downstream applications like investigating protein-related diseases and guiding rational protein design.

## 1 INTRODUCTION

Feature representation is an important aspect that contributes to the success of machine-learning methods, which are primarily data-driven. One attempts to translate the domain knowledge of a given learning task by defining features that are relevant and contribute to the learning task at hand. Also, this depends on the nature of the available dataset. Use of protein language models (PLMs) trained on large unlabeled datasets has become commonplace in computational biology (Pokharel et al., 2025; Weissenow & Rost, 2025). A common usage of PLMs is to get feature representation of protein sequences, which are then used to train a simpler machine learning model for a downstream prediction task. The PLM-based representations are high-dimensional and achieve high predictive performance on a wide range of tasks, which can be further improved with minimal fine-tuning (Weissenow & Rost, 2025). However, the uninterpretability of the PLM-based representations and the inherent complexity of the PLM pose as barriers towards gaining intelligible and actionable insights into the input (protein-sequence) and output (prediction) relationship, so as to extend domain knowledge.

In this work, we invest in hand-crafted feature engineering from protein datasets and explore how well these interpretable features fare against uninterpretable PLM-based features in predictive performance. We focus on the task of classifying protein domains into their CATH (Sillitoe et al., 2020) superfamilies by using different types of features computed from the protein's sequence and structure.

Proteins are often segmented into domains that are subunits of a protein structure that fold independently of the rest of the structure (Kolodny et al., 2013a). Various studies estimate that the number of folds adopted by proteins in nature is between 1,000 and 10,000 (Kolodny et al., 2013b). CATH is a database that groups protein domains identified from PDB (Protein Data Bank) structures into hierarchical groups based on the similarity of their 3-dimensional fold. The protein domains in the CATH database are classified into 6,631 homologous superfamilies. Each homologous superfamily in CATH is a group of protein domains, with each superfamily containing domains having structures similar to each other.

In this work, we compute different types of feature vectors from protein sequence/structure, and evaluate how well each type of feature vector can distinguish a given CATH superfamily from all others. We train classifiers to predict the CATH homologous superfamily of a protein domain using 9 different types of feature vectors. We do a robust study on 45 superfamilies curated based on the number of available sequences. This dataset of 45 superfamilies is diverse, based on various aspects (discussed in Section 3.1). A one-vs-all SVM classifier (with loss function capable of handling class imbalance) is trained using each type of feature vector to predict the CATH homologous superfamily of a protein domain. These feature vectors can be categorised in two ways,

- sequence-based vs structure-based, and

- hand-crafted (interpretable) vs protein-language-model (PLM) based (non-interpretable)

The nine feature vector types capture information at varying levels of granularity (coarse-grained to fine-grained) from the protein's sequence/structure. The motivation here is to identify which type of feature vector, and thereby which level of information, is effective in distinguishing CATH superfamilies. This can be useful in a downstream task to identify features that are characteristic of a CATH superfamily using feature importance measures. This is especially useful if interpretable sequence-based features can be identified that are characteristic of a CATH superfamily. As these can be used in designing new proteins that are required to have a structure as characterised by a CATH superfamily.

The main contributions of this work are as follows,

- We highlight a trade-off between the predictive performance and interpretability of input features, using PLM-based features and hand-crafted features. The PLM-based features have high predictive performance but low/no interpretability, while hand-crafted features have relatively lower performance but high interpretability.

- We propose two novel structure-based feature engineerings: OCPC (ordered contact pair composition) and CSIC (contact separation interval composition). The features and dimensions of CSIC are determined by the distribution of the contact sequence separation of a given superfamily, for which a one-vs-all classification is performed.

- We propose a novel sequence feature engineering $k$OAAC ($k$-ordered amino acid composition), that encodes increasing levels of sequence order information with higher values of $k$

- We propose a new feature engineering from the attention matrix of PLM: ProtBERT-Attn. This aggregates attention values based on amino-acid types.

- Despite significant class imbalance in the one-vs-all classification of superfamilies, we see high predictive performance with structure-based feature CSIC.

- We find that CSIC, due to its low overfitting, can perform comparably with the PLM-based feature, ProtBERT-Attn. This is significant, as CSIC features are interpretable. Moreover, with the advent of Alphafold (Jumper et al., 2021), quality structure predictions are available that can be utilised for computing the CSIC feature.

We discuss our feature engineering in detail in the next section (Section 2). We discuss the details on the dataset used in Section 3 and the methodology for training/evaluation of classifiers in Section 4. We discuss the results of our computational experiments in Section 5. Conclusions and future directions for work are discussed in Section 6.

## 2 FEATURE ENGINEERING

We use broadly three types of feature vectors engineered from the protein domain, *which encode different levels of information from the protein domain.* For computing these features, we use the information of the types of amino acids that constitute the protein. We consider the standard 20 amino acid types, (A, R, N, D, C, E, Q, G, H, I, L, K, M, F, P, S, T, W, Y, V), we refer to these using $\mathcal{T} = \{t_1, t_2, \cdots, t_{20}\}$.

Table 1: A summary of the types of feature vectors computed from a protein domain's sequence or structure for training one-vs-all classifiers.

| Feature Engineering | Dimension | Feature vector component description/interpretation |
|---|---|---|
| *Hand-crafted, from sequence* | | |
| amino acid composition (AAC) | 20 | The number of times an amino acid type $t_i$ occurs in the sequence. Each dimension corresponds to a different amino acid type. |
| Dipeptide composition (DPC) | $20^2 = 400$ | The number of times amino acid type pairs $(t_{i_1}, t_{i_2})$ occur adjacent to each other in the sequence in that order. Each dimension corresponds to a different ordered-pair of amino acid types. |
| Tripeptide composition (TPC) | $20^3 = 8000$ | The number of times amino acid types $(t_{i_1}, t_{i_2}, t_{i_3})$ occur adjacent to each other in the sequence in that respective order. Each dimension corresponds to a different ordered triplet of amino acid types. |
| 2-ordered amino acid composition (2OAAC) | $20^2 = 400$ | Out of the $\binom{L}{2}$ ordered position pairs $(p_{j_1}, p_{j_2}), j_1 < j_2$, in the sequence, the number of such position pairs having amino acid types $(t_{i_1}, t_{i_2})$ in that respective order. Each dimension corresponds to a different ordered-pair of amino acid types. |
| 3-ordered amino acid composition (3OAAC) | $20^3 = 8000$ | Out of the $\binom{L}{3}$ ordered position triplets $(p_{j_1}, p_{j_2}, p_{j_3}), j_1 < j_2 < j_3$, in the sequence, the number of position triplets having amino acid types $(t_{i_1}, t_{i_2}, t_{i_3})$ in that respective order. Each dimension corresponds to a different ordered-triplet of amino acid types. |
| *Hand-crafted, from structure* | | |
| Ordered contact pairs composition (OCPC) | $20^2 = 400$ | The number of times the amino acid type pair $(t_{i_1}, t_{i_2})$ are in contact in the structure and occurs in the sequence in the same relative order. Each dimension corresponds to a different ordered pair of amino acid types. |
| Contact separation interval composition (CSIC) | $K \times 20$ ($K$ is determined from the data) | The number of contacts an amino acid type $t_i$ has in the structure with another amino acid separated by at least $l$ and at most $u$ residues in the sequence. Each dimension corresponds to a different amino acid type $t_i$ and interval $(l, u)$ combination. $K$ is the number of such intervals considered. |
| *Protein language model (PLM) based, from sequence* | | |
| ProtBERT-Emb | 1024 | Averaged embeddings of the final layer of protein language model ProtBERT. No interpretation for dimensions. |
| ProtBERT-Attn | $16 \times 20 = 320$ | Each dimension is the aggregation of the row-sum of the attention-matrix for the rows corresponding to amino-type $t_1$. This is done for each attention-head (total 16). The attention matrix is from the final layer of ProtBERT. No interpretation for attention-values. |

### 2.1 HAND-CRAFTED FEATURES FROM SEQUENCE

From the sequence, we compute one type of feature vector that doesn't utilise any sequence order information and four other types of feature vectors that encode varying levels of sequence order information. These are discussed below.

#### 2.1.1 AMINO ACID COMPOSITION (AAC)

As a simplistic feature, we count the number of occurrences of each of the 20 amino acid types $\mathcal{T}$, as defined in Section 2 (para 1). This results in a 20-dimensional feature vector. For a protein sequence $\mathbf{p} = (p_1, p_2, \ldots, p_L)$ of length $L$ with $p_j \in \mathcal{T}$ being one of the standard 20 amino acids, the AAC

feature $\mathbf{x}^{AAC} \in \mathbb{R}^{20}$ for $\mathbf{p}$ is computed as follows, $x_i^{AAC} = \sum_{j=1}^{L} \mathbf{1}_{\{P_j = t_i\}}, \forall i \in \{1, 2, \cdots, 20\}$. Here, $t_i \in \mathcal{T}$ is one of the defined amino acid types.

The AAC feature completely ignores the amino acids' order in the sequence. Two protein sequences $\mathbf{p}$ and $\mathbf{q}$ will have the same amino acid composition if $\mathbf{q}$ is a permutation of $\mathbf{p}$. Thus, we introduce the $k$-ordered amino acid composition ($k$OAAC) feature vector, which considers the amino acids' relative order in the sequence. We discuss this in detail below.

### 2.1.2 Features that encode sequence order

We use 4 types of features that encode sequence order information, partially, into the feature vector dimensions by accounting for the relative order/position of amino acids in the protein sequence. These are *di-peptide composition (DPC), tri-peptide composition (TPC), 2-ordered amino acid composition (2OAAC)* and *3-ordered amino acid composition (3OAAC)*. DPC and TPC are existing and widely used features, while 2OAAC and 3OAAC are novel feature engineerings that are introduced in this work.

DPC is a ($20^2 =$) 400-dimensional feature that computes the count of the contiguous 2-mers of given amino acid types in the sequence. Similarly, TPC is a ($20^3 =$) 8000-dimensional feature that computes the count of contiguous 3-mers of given amino acid types in the sequence. In general for $k$-peptide composition ($k$PC), the count of the occurrence of a $k$-mer $(t_{i_1}, t_{i_2}, \cdots, t_{i_k})$ of amino acid types, corresponding to feature dimension $i$, in a sequence $\mathbf{p}$ is given as,

$$x_i^{kPC} = x_{(i_1, i_2, \cdots, i_k)}^{kPC}, \qquad i = i_1 + \sum_{r=2}^{k} 20^r (i_r - 1) \in [20^k]$$

$$= \sum_{1 \leq j \leq L-k+1} \mathbf{1}_{\{P_j = t_{i_1}, P_{j+1} = t_{i_2}, \cdots, P_{j+k-1} = t_{i_k}\}} \qquad (1)$$

We also introduce two novel features that encode sequence order information, 2OAAC and 3OAAC.

2OAAC is similar to DPC but allows any number of residues (can be even 0) between the two amino acids, with the order of the two amino acids maintained (i.e., K_M is distinct from M_K). Consider an example sequence 'M R K P M M W A E L R V'. The ordered pair (M, R) occurs 4 times at positions $(1, 2), (1, 11), (5, 11)$, and $(6, 11)$. Meanwhile, the ordered pair (R, M) occurs twice at positions $(2, 5)$ and $(2, 6)$. Similarly, the $20^2 = 400$ dimensional 2OAAC feature vector can be computed by counting the occurrence of all $20^2$ ordered pairs of amino acids.

Likewise, 3OAAC is similar to TPC but allows any number of residues (can be even 0) between the three amino acids, with the order of the three amino acids maintained. Figure 1 illustrates how a $20^3 = 8000$ dimensional 3OAAC feature is computed.

Figure 1: For $k = 3$ in feature description in Equation (2), two occurrences of the ordered tuple (K,M,R) in a sequence of length 12. Similarly, the $20^3 = 8000$ dimensional 3OAAC feature vector for the sequence can be computed by counting the occurrence of all $20^3$ ordered 3-tuples of amino acids.

In general, for $k$OAAC, feature dimension $i$ corresponding to the ordered tuple $(t_{i_1}, t_{i_2}, \cdots, t_{i_k})$ for a sequence $\mathbf{p}$ can be computed as,

$$x_i^{kOAAC} = x_{(i_1, i_2, \cdots, i_k)}^{kOAAC}, \qquad i = i_1 + \sum_{r=2}^{k} 20^r (i_r - 1) \in [20^k]$$

$$= \sum_{1 \leq j_1 < j_2 < \cdots < j_k \leq L} \mathbf{1}_{\{P_j = t_{i_1}, P_{j+1} = t_{i_2}, \cdots, P_{j+k-1} = t_{i_k}\}} \qquad (2)$$

A brute-force counting of the occurrence of $(t_{i_1}, t_{i_2}, \cdots, t_{i_k})$ in the sequence $\mathbf{p}$ will have a computational complexity of $\mathcal{O}(\binom{L}{k})$. While using dynamic programming, it can be done in $\mathcal{O}(L)$. However, the space complexity for this feature computation is $\mathcal{O}(20^k)$.

**$k$PC looses AAC information while $k$OAAC retains it** $k$PC encodes some sequence order information, but the AAC information cannot be recovered from it. This can be illustrated using a simple example. Consider the two sequences 'AARRA' and 'RRAAR'. Both the sequences have the same DPC, {'AA':1, 'AR':1, 'RR':1, 'RA':1}, while their AACs are different {'A':3, 'R':2} and {'A':2, 'R':3}. Thus, two sequences with the same $k$PC may not have the same AAC. However, if two sequences have the same $k$OAAC, then they will have the same AAC. For example, the AAC for a sequence can be recovered from its 2OAAC as follows,

$$ x_{i_1}^{AAC} = \frac{1}{L-1} \sum_{i_2 \in [20]} \left( x_{(i_1, i_2)}^{2OAAC} + x_{(i_2, i_1)}^{2OAAC} \right) \qquad (3) $$

In general, the $[k-1]$OAAC feature vector of a sequence can be recovered from its $k$OAAC feature vector. Thus, two sequence with same $k$OAAC will have the same $[k-1]$OAAC feature vector. However, two features with the same $k$PC may not have the same $[k-1]$PC.

## 2.2 HAND-CRAFTED FEATURES FROM STRUCTURE

We propose two types of *novel feature vectors* from the 3-dimensional structure of the protein domains. Availability of high-accuracy predicted 3D structures of proteins makes it possible to compute these vectors. In particular, Alphafold has provided high-accuracy 3D strucures for most proteins, which makes it possible to compute feature vectors that we are proposing here. For computing these features, we first compute a contact map from the protein's structure. For a protein $\mathbf{p}$ with sequence length $L$, the contact map $C$ is a square matrix of the form $C \in \{0, 1\}^{L \times L}$. Where $C_{j,k} = 1$ if the distance between the centroids of the $j^{th}$ and $k^{th}$ amino acids is less than a given threshold $\theta$ in the 3D structure. We use $\theta = 7\text{Å}$ (angstroms). The size of $C$ depends on the protein sequence length. We use the contact map of protein domain to compute the two types of strucure-based feature vectors that are dicussed below.

### 2.2.1 ORDERED CONTACT PAIRS COMPOSITION (OCPC)

We define OCPC as a $(20^2 =)$ 400-dimensional feature that computes the count of contacts formed by given pairs of amino acid types in the protein structure. Here, the contacts are defined by the contact map. The relative order in which the two amino acids defining the contact occur in the sequence is also considered. The OCPC feature dimension $i$ for the amino acid type pair $(t_{i_1}, t_{i_2})$ from protein $\mathbf{p}$ with its contact map $C$ is computed as follows,

$$ x_i^{OCPC} = \sum_{1 \le j_1 < j_2 \le L} \mathbf{1}_{\{p_{j_1} = t_{i_1}, p_{j_2} = t_{i_2}\}} \times C_{j_1, j_2}, \qquad i = i_1 + 20(i_2 - 1) \in [20^2] \qquad (4) $$

Thus, the feature dimensions of OCPC contain two kinds of information. One is the amino acid type pairs that are in contact in the 3-dimensional structure of the protein, and the other is the relative order in which these contact-forming amino acid pairs occur in the sequence.

### 2.2.2 CONTACT SEPARATION INTERVAL COMPOSITION (CSIC)

We define CSIC as $K \times 20$ dimensional feature that counts the number of contacts a given amino acid type has with any other amino acid that is within a given sequence separation range. Here, $K$ is the number of sequence separation intervals/ranges defined by the user. Let the $K$ intervals defined by the user be, $\mathcal{I} = \{[l_1, u_1], [l_2, u_2], \cdots, [l_K, u_K]\}$. The CSIC feature dimension $i$ for the amino acid type $t_{i_1}$ and interval $[l_k, u_k]$ from protein $\mathbf{p}$ with its contact map $C$ is computed as follows,

$$x_i^{CSIC} = x_{i_1,(l_k,u_k)}^{CSIC}, \qquad i = i_1 + 20(k-1) \in [K \times 20]$$

$$= \sum_{1 \le j_1 < j_2 \le L} C_{j_1,j_2} \times \mathbf{1}_{\{l_k \le j_2 - j_1 \le u_k\}} \times \mathbf{1}_{\{p_{j_1} = t_{i_1} \vee p_{j_2} = t_{i_1}\}} \tag{5}$$

As in OCPC, the feature dimensions of CSIC contain two kinds of information. One is the number of contacts an amino acid type forms with other amino acids in the 3-dimensional structure of the protein. The other is how separated in the sequence are these amino acids that form contacts.

## 2.3 PROTEIN LANGUAGE MODEL (PLM) BASED FEATURES FROM SEQUENCE

Using ProtBERT (Elnaggar et al., 2021), a pre-trained PLM, we compute two types of feature vectors from it.

Given an input protein sequence $\mathbf{p}$ of length $L$, ProtBERT returns $L$ number of 1024-dimensional embedding vectors corresponding to each position of the input sequence. This can be viewed as a $L \times 1024$ matrix. We take the average of this matrix along the sequence length dimension to get a single 1024-dimensional embedding vector for the input sequence $\mathbf{p}$. We refer to this feature vector type as ProtBERT-Emb. The feature dimensions of ProtBERT-Emb have no domain-knowledge-based interpretable notion.

Another feature vector that we compute from ProtBERT is using the attention-matrix from its final layer. Each layer of ProtBERT has 16 attention-heads, each generating an attention-matrix. Let's refer to attention-matrix from the final layer's $h^{th}$ attention-head as $A^h$ for the input sequence $\mathbf{p}$. $A^h$ is an $L \times L$ column stochastic matrix, i.e. $\sum_{i=1}^{l} A_{i,j}^h = 1$. We compute a $(16 \times 20 =)$ 320-dimensional feature matrix that aggregates the attention values according to amino acid types. We refer to this feature vector type as ProtBERT-Attn. The ProtBERT-Attn feature dimension $i$ for amino acid type $t_{i_1}$ and attention-head $h$ is computed as follows,

$$x_i^{\text{ProtBERT-Attn}} = x_{(i_1,h)}^{\text{ProtBERT-Attn}}, \qquad i = i_1 + 20(h-1) \in [16 \times 20]$$

$$= \sum_{j_1=1}^{L} \left( \sum_{j_2=1}^{L} A_{j_1,j_2}^h \right) \times \mathbf{1}_{\{p_i = t_{i_1}\}} \tag{6}$$

Although the feature dimension for ProtBERT-Attn is defined by amino acid types, the attention values aggregated do not have a domain-knowledge-based interpretation.

## 3 DATASETS AND THEIR KEY CHARACTERISTICS

The CATH database (Sillitoe et al., 2020) categorises more than 0.5 million protein domains at four hierarchical classification levels: Class → Architecture → Topology → Homologous superfamily. Class is based on the predominant secondary structure component in the fold. Architecture is based on the relative arrangement of secondary structures in the 3-dimensional space. Topology is based on how the secondary structure components are connected in a fold. Homologous superfamily is based on evidence of common ancestry

The protein domains in the CATH database are classified into 6,631 homologous superfamilies. We use non-redundant datasets of homologous superfamilies with 35% as the sequence identity threshold (downloaded from the CATH website); in all, this includes 32,388 CATH domains. We train a binary classifier for each superfamily that has at least 100 representative domain sequences, i.e., for 45 superfamilies. Figure 2 shows the number of representative domains and the sequence-length distribution for each of the 45 superfamilies.

### 3.1 DATASET DIVERSITY

We find that the selected datasets for 45 superfamilies are very diverse. More details are in Appendix A.1. We highlight the important aspects here:

- *Sequence diversity*: No two sequences have more than 35% sequence identity
- *Structural diversity*: Of the 45 superfamilies 6, 11, and 28 belong to 'mainly alpha' (i.e. CATH ID 1.*), 'mainly beta' (i.e. CATH ID 2.*) and 'alpha beta' (i.e. CATH ID 3.*) classes in the 1st level of CATH hierarchical classification (Figure 2). Alpha and beta denote secondary structure patterns. 'Mainly alpha' have primarily alpha helices, 'mainly beta' have primarily beta strands, and 'alpha beta' are a mixture of both.
- The dataset size for a given superfamily varies from 100 to 873 sequences (Figure 2).
- There is no correlation between the variation of sequence lengths and the number of representative CATH domain sequences of a given superfamily.

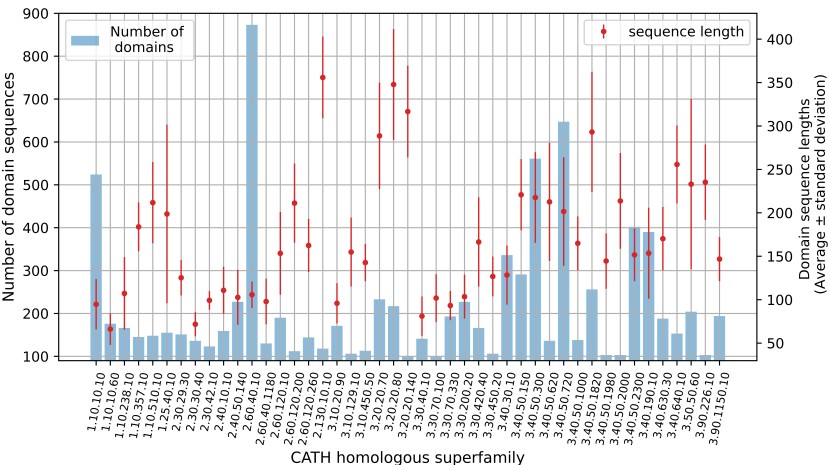

Figure 2: The bar plot shows the number of representative domain sequences (left y-axis) available for 45 CATH homologous superfamilies in the non-redundant dataset with a 35% sequence identity threshold. The scatter plot with error bars shows the distribution of the length of the domain sequences (right y-axis) for each superfamily. The 45 superfamilies are selected from a total of 6,631 CATH superfamilies based on the criteria that they have at least 100 representative domain sequences after applying the redundancy filter.

## 4 CLASSIFIERS FOR CATH SUPERFAMILY PREDICTION

We train one-vs-all classifiers to predict the CATH homologous superfamily of a protein domain.

### 4.1 ONE-VS-ALL LINEAR SVM CLASSIFIERS

For each of the selected 45 superfamilies described in Section 3, we train a one-vs-all binary classifier predicting whether a given domain sequence belongs to the corresponding superfamily or any of the other 6,630 CATH homologous superfamilies. For each classifier, the positive class dataset comprises protein domain sequences from one of the 45 superfamilies, and the negative class dataset comprises domain sequences from all the other 6,630 superfamilies. The train and test set for a classifier is made with an 80:20 split of both positive and negative datasets.

#### 4.1.1 TRAINING AND EVALUATION WITH CLASS-IMBALANCE

If there are $m_1$ samples in the positive dataset, then the negative dataset has $m_2 = 32,388 - m_1$. We have $m_1 \in [100, 873]$ across the selected datasets (Figure 2), therefore the range of class-imbalance ratio is $m_1/m_2 \in [0.003, 0.028]$. Thus, each classifier's train/test data has a large class imbalance (an average imbalance of 1:197). To account for this, the test performance of a classifier was measured using the Arithmetic Mean (AM) of specificity and sensitivity (Brodersen et al., 2010). Also, an empirically class-balanced version of squared hinge loss is used in training the SVM as suggested in Menon et al. (2013) for statistical consistency with the AM score. For each classifier, 10% of the

train set is used as a validation set for tuning the SVM regularisation hyperparameter $C$. $C$ is inversely proportional to the strength of regularisation. The average AM scores are reported with 5 random train/test splits for each superfamily with different features.

Scikit-learn's (Pedregosa et al., 2011) `LinearSVC` module is used for training the classifiers for all features except TPC and 3OAAC. As TPC and 3OAAC features are very high-dimensional, we used scikit-learn's `SGDClassifier` module with hinge-loss and mini-batch size of 10,000 for training the linear classifiers using these features.

### 4.1.2 CSIC INTERVALS FOR ONE-VS-ALL CATH SUPERFAMILY CLASSIFICATION

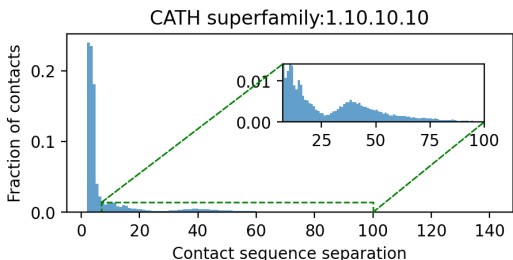

Figure 3: The plot shows the distribution of the contact length separations of all the contacts from the structures of CATH superfamily 1.10.10.10

Recall from Section 2.2.2 that for CSIC feature computation, a user-defined input, i.e. a set of sequence separation intervals $\mathcal{I}$ is required. We define this $\mathcal{I}$ in a data-driven manner based on the superfamily for which the one-vs-all classifier is trained. For doing say, superfamily '1.10.10.10'-vs-'other' classification, we first look at the distribution of the sequence separation of the contact residues of all the structures of this family. See Figure 3. Contacts by residues that are adjacent in the sequence are ignored. We see the distribution is light-tailed with the highest concentration at 2. Zooming in on the tail, we see that the distribution has many small modes. To infer the multiple prominent modes in the distribution, we approximate it using Gaussian mixtures (Bishop, 2006). From each of the fitted Gaussians, we compute the 75% confidence interval with equal areas around the median. Thus, if $K$ Gaussians were fitted, we get $K$ intervals which we use as $\mathcal{I}$. We refer to this feature, where CSIC intervals are computed using Gaussian mixtures, as CSIC-Gaussian. Since the contact separation distribution has a semi-infinite support, we also use gamma mixtures (Xiong et al., 2024) for defining $\mathcal{I}$. We refer to this feature as CSIC-Gamma. The value of $K$ thus depends on the superfamily for which the one-vs-all classification is done. In our experiments, $K$ takes values from 4 to 13.

In the next section, we discuss the performance of the classifier using the different features.

## 5 RESULTS AND DISCUSSION

Table 2: The average train/test AM scores over 5 random splits are again averaged across the 45 superfamilies. Similarly, the standard deviations (s.d.) of train/test AM scores over 5 random splits are again averaged across the 45 superfamilies.

| AM | Avg. | Hand-crafted sequence-based | | | | | Hand-crafted structure-based | | | PLM-based | |
|---|---|---|---|---|---|---|---|---|---|---|---|
| | | AAC | DPC | 2OAAC | TPC | 3OAAC | OCPC | CSIC-Gauss | CSIC-Gamm | PB-Attn | PB-Emb |
| Train | Avg. | 81.8 | 96.1 | 92.2 | 93.1 | 83.4 | 94.7 | 96.8 | 96.7 | 97.1 | 99.7 |
| | (s.d.) | (0.9) | (1.9) | (3.7) | (5.3) | (4.8) | (1.8) | (1.5) | (1.3) | (1.9) | (0.2) |
| Test | Avg. | 79.9 | 74.3 | 78.7 | 77.7 | 77.7 | 79.2 | 88.8 | 88.4 | 88.5 | 96.5 |
| | (s.d.) | (2.4) | (6.8) | (4.8) | (5.0) | (4.0) | (4.6) | (4.4) | (3.7) | (3.8) | (2.0) |

The one-vs-all classification scores using the 10 different feature vectors across the 45 CATH superfamilies are reported in Section 5 and Table 2. We summarise our main observations below:

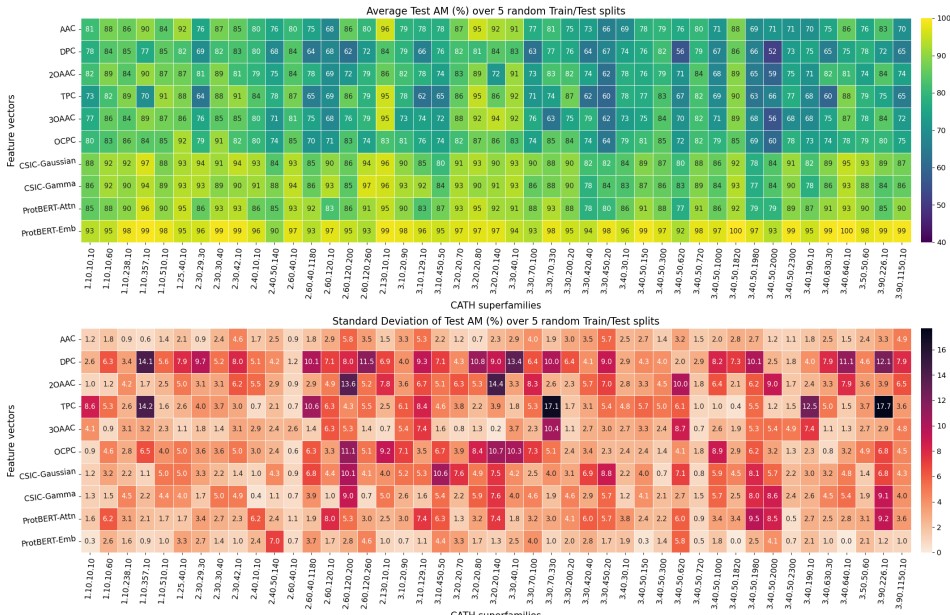

Figure 4: Average and standard deviations of Test AM scores (in the form of heatmaps) across 5 random train/test splits for the 45 superfamilies using each of the 10 feature vectors.

- Amongst all the 10 different feature vectors considered, the PLM-based feature ProtBERT-Emb has the highest predictive performance ($> 90\%$) across the 45 superfamilies. The standard deviation of test scores across random splits is also the least for ProtBERT-Emb.

- AAC feature, which does not use any sequence order information, has $>60\%$ test AM scores across the 45 superfamilies. The test AM is $> 80\%$ for 19 superfamilies and $> 90\%$ for 6 superfamilies. The standard deviation of test scores across random splits is also low.

- All hand-crafted features except AAC have high standard deviations of test scores across random splits. The DPC, 2OAAC, TPC, and OCPC features also have significant overfitting, as can be seen from the difference between train and test AM scores.

- Hand-crafted structure-based CSIC features, with low overfitting, have performance comparable with ProtBERT-Attn; the average test scores across superfamilies being $\approx 88\%$.

## 6 CONCLUSIONS

PLM-based features outperform hand-crafted features in the CATH superfamily classification task. As ProtBERT is pre-trained exclusively using protein sequences, the high classification performances using ProtBERT-Emb suggest that there could be sequence features that are characteristic of a CATH superfamily. However, we don't know what these characteristic features are as the individual components of the feature vector are not interpretable.

In contrast, hand-crafted features are highly interpretable, even though they do not achieve as high classification scores as PLM features. However, we find that carefully crafted features such as CSIC can have predictive performance that is comparable to some PLM-based features like ProtBERT-Attn. It strikes a balance between performance and interpretability and has low overfitting. Such features could be useful in inferring features that are characteristic of a CATH superfamily. The CSIC feature components are rich in information about amino acid types forming contacts and the sequence separation at which the contacts are formed.

We emphasize that the hand-crafted structure-based feature engineering offers a template for engineering similar features for other protein related classification tasks. This may need a suitable combination of domain knowledge and statistical techniques, similar to the use of contact sequence separation distribution in CSIC.

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

# A  APPENDIX

## A.1  DATASET DIVERSITY

We find that the selected datasets for 45 superfamilies are very diverse. This is due to the following reasons:

- No two domain sequences, whether from the same or different superfamilies, have more than 35% sequence identity.

- Of the 45 superfamilies 6, 11, and 28 belong to 'mainly alpha' (i.e. CATH ID 1.*), 'mainly beta' (i.e. CATH ID 2.*) and 'alpha beta' (i.e. CATH ID 3.*) classes in the 1st level of CATH hierarchical classification (Figure 2). Alpha and beta denote secondary structure patterns. 'Mainly alpha' have primarily alpha helices, 'mainly beta' have primarily beta strands, and 'alpha beta' are a mixture of both.

- The dataset size for a given superfamily varies from 100 sequences to 873 sequences (Figure 2).

- The length distribution of the domain sequences varies between superfamilies (Figure 2). For example, the sequence lengths of CATH IDs 2.130.10.10 and 3.20.20.80 are $356 \pm 47$ and $348 \pm 64$ amino acids, respectively, while those of 1.10.10.60 and 2.30.30.40 are $66 \pm 18$ and $72 \pm 14$ amino acids, respectively.

- The variations of sequence lengths within some superfamilies are much higher than those for others. For example, for CATH IDs 1.25.40.10 and 3.50.50.60, the standard deviation of the sequence lengths is 103 and 98 amino acids, respectively. Meanwhile, for CATH IDs 2.30.30.40 and 2.30.42.10, the standard deviation of the sequence lengths is only 14 and 11 amino acids, respectively.

- There is no correlation between the variation of sequence lengths and the number of representative CATH domain sequences of a given superfamily.

