# OpenReview forum: "Performance vs interpretability trade-off of hand-crafted and language model features: The case of protein superfamily classification"
_ICLR.cc/2026/Conference — Submitted to ICLR 2026_

### Official Review · Reviewer_qdz7 · 2025-10-17

**Soundness:** 3
**Presentation:** 2
**Contribution:** 1
**Rating:** 2
**Confidence:** 4

**Summary:**

In their submission, the authors investigate different feature sets for a particular protein classification task, namely protein superfamily classification. The authors carefully introduce three different feature sets: sequence-based (n-gram), structure-based (based on contact pairs derived from protein structure), protein foundation model features (attention or embeddings from ProtBERT). The structure-based feature were proposed by the authors and they stress their broad availability by leveraging AlphaFold structure. They train SVM classifiers on different feature sets in a one vs. all fashion to discriminate most informative feature sets. Protein Language Model embeddings turn out to be most predictive, but not interpretable as the authors point out correctly, closely followed by contact-based features proposed by the authors.

**Strengths:**

* Pedagogical introduction to the field and to the different feature sets in a way that should be accessible also for people unfamiliar with a computational biology context.
* Newly proposed structural features that are apparently very discriminative (at least for the task at hand) that can be derived from widely available structural information (e.g. from AlphaFold)
* The authors correctly emphasize the tension between the discriminative power of features and their interpretability, which is important for applications in the Natural Sciences
* The authors use a sensible way of fixing the distance threshold for their contact-based features in a data-dependent manner.

**Weaknesses:**

* Just a single task is considered. More general insights could be drawn if further qualitatively different tasks would be considered e.g. enzyme classification or gene ontology prediction.
* The authors miss MSA-features as important and very powerful category of features.
* No combination of feature sets is considered. It would also be interesting to understand their individual contributions (e.g. using a Shapley formalism) to quantify which features contributes how much and potential overlap between different feature sets.
* There is a strong imbalance between setup and results. The actual results constitute barely one page in the manuscript. The largest part of the paper provides feature definitions (which could to some degree also be provided in the supplementary material). This would free up some space to add more details on experiments and implications.
* No statements about the statistical significance of their findings were made. These could for example be implemented via emprical bootstrapping the performance difference on the test set.
* Without doubt, the proposed structural features are more interpretable than foundation model features but are also not as interpretable as for example the sequence-based features. To turn this into a stronger submission, it would be nice to see some more specific insights from these more interpretable features to get an idea what kind of insights they could enable.

**Questions:**

* Did the authors explore any other protein language models, e.g. ESM2?
* Can the authors motivate their choice of classification algorithm?
* Was the structural information taken from experimental data or from AlphaFold? It would be interesting to quantify the difference between the two to assess whether the author's scalability argument (by using AlphaFold structures) is valid.

---

> ### Author Response · Authors · 2025-11-21
>
> *We thank the reviewer for taking the time to review our submission. Please find below our answers to the questions asked*
>
> # Questions
>
> ## Did we explore other PLMs?
> No, We did not explore other PLMs. This is because ProtBERT achieves an average test score of 96.5\% across the 45 superfamilies. This is already a high score, and using any other PLM can achieve rather minor - 4.5\% - improvement on this. Thus, *ProtBERT sufficiently demonstrates the high predictive power of PLMs*. Furthermore, the *notion of interpretability is largely similar across PLMs*, such as ESM and ProtT5. That is by finding correlations between attention values and known protein properties ([Simon \& Zou, Nature Methods 2025](https://www.nature.com/articles/s41592-025-02836-7), [Vig et al., ICLR 2021](https://openreview.net/forum?id=YWtLZvLmud7)).
>
> Thus, we believe that ProtBERT adequately represents the PLMs on both performance and interpretability objectives. Hence, other PLMs were not used in this study as the focus of the work is towards engineering interpretable features tailored for this task, which can achieve competitive performance.
>
>
> ## Motivation for choice of classification algorithm
>
> Using each of the features, we train a Linear SVM classifier with a class-balanced hinge loss. The class-balanced hinge loss is statistically consistent with the AM score ([Menon et al., ICML 2013](https://proceedings.mlr.press/v28/menon13a.html)) that is used for performance evaluation.
>
> *We use a linear SVM classifier for interpretability*. All the hand-crafted features have positive counts of interpretable quantities present in the protein sequence/structure (Please see Table 1 of the main paper). The *sign of the one-vs-all linear SVM coefficients* can be used to infer which group of features is characteristically rich (or deficient) in a given CATH superfamily. This is an important insight we can have; we provide more details in the response *`What insight can be gained?'*.
>
>
>
> ## Was the structural information from experimental data or Alphafold? Is the scalability argument using Alphafold structures valid?
>
> The structural information used in our work is from experimental data (from the [Protein Data Bank](https://www.rcsb.org/)), which is for computing OCPC and CSIC features.
>
> **Scalability using Alphafold.**  Since experimentally determined structures are only available for limited protein sequences,  we want to emphasize that this not a bottleneck for CSIC/OCPC feature computation as quality structure predictions are now available via Alphafold.
>
> *(Our responses to weaknesses are continued in the next two comments)*

---

> ### Author Response · Authors · 2025-11-21
>
> # Weaknesses
>
> ## Single task is considered
>
> We thank the Reviewer for this suggestion of other possible learning tasks. We didn't pursue these for the reasons that we elaborate.
>
> Unlike PLMs, which are considered foundation models that can adapt to a wide range of downstream tasks, we focus on interpretable hand-crafted feature engineering tailored for a specific task. For this, we consider the task of CATH superfamily classification of 45 superfamilies in our experiments. Solving this task in an interpretable manner can be useful in identifying features that are characteristic of a CATH superfamily's fold.
>
> **Why not enzyme classification or gene ontology prediction (GO)?**. Enzyme classification is primarily done via predicting the [enzyme commission (EC)](https://en.wikipedia.org/wiki/Enzyme_Commission_number) number of an enzyme. We do not consider EC number of GO annotation prediction, as many proteins with no common sequence or structural features have the same EC number or GO annotations. Examples of proteins with different structure but same function after convergent evolution can be seen here: [link](https://en.wikipedia.org/wiki/List_of_examples_of_convergent_evolution#In_proteins,_enzymes_and_biochemical_pathways). GO annotations like cellular-component can be same for very different proteins that function at the same cellular location. Thus, we did not consider EC number or GO prediction tasks, as *there may be no characteristic sequence/structural features for the proteins with a given label*.
>
>
> ## Missing MSA features
>
> MSA is not applicable to the sequence datasets in our experiments. This is because no two sequences in the dataset have more than 35\% sequence similarity.
> The following file: [link](https://anonymous.4open.science/r/cath_classification-8A0C/clustalo-cath_1_10_10_10.fa), contains MSA computed for CATH superfamily 1.10.10.10 using alignment tool from [ebi.ac.uk](https://www.ebi.ac.uk/jdispatcher/msa/clustalo). We find that for sequences within the family itself, the alignment has *a large number of gaps with very little or no conservation at any position*. Thus, the MSA is not a useful feature in this case.
>
>
> ## No feature combinations considered
>
> As mentioned in line 463, page 9 in the main paper, we observe that with the hand-crafted features DPC, 2OAAC, TPC and 3OAAC, there is significant model overfitting. We observed *increased overfitting* when combining these features. When combining AAC and CSIC features (the hand-crafted features with the least overfitting), we do not see a significant difference in test scores.
>
> ## A large part of the paper is feature definitions. The actual results are only one page.
>
> Since we focused on hand-crafted features with an eye on interpretable ones, we have devoted a significant discussion to them. Yes, we do agree that the results were compressed into a single page. We will move some of the discussion to the appendix. We will include more details suggested by the Reviewer and details of the additional experiments in the main paper, utilising the space provided by moving some feature definition details to the appendix. We thank the Reviewer for this useful suggestion.
>
> ## No statement about the statistical significance of findings
>
> As already discussed in Section 4.1.1, the experiments were repeated on 5 different train/test random splits. The mean and standard deviations of the test scores across these random splits have already been reported in Figure 4 of the main paper.
>
>
> *(Response to another weakness is in the next comment)*

---

> ### Author Response · Authors · 2025-11-21
>
> # Weaknesses
>
> ## Structural features not as interpretable as sequence features. What insights can they enable?
>
> We respectfully disagree with the reviewer's comment that the structural features are less interpretable than sequence features. The CSIC structural features are very rich in information. As already described in Table 1, a feature component of CSIC is interpreted as follows: *'number of contacts an amino acid type $t_i$ has in the structure with another amino acid separated by at least $l$ and at most $u$ residues in the sequence'*.
> Considering an interval of sequence separation instead of singleton value makes it robust to [indel](https://en.wikipedia.org/wiki/Indel) mutations. The intervals $(l,u)$ are determined in a data-driven manner using both Gaussian and gamma mixture models and both are reported (Results Section of the main paper).
>
> **What insight can be gained?** All the hand-crafted features are positive counts of interpretable quantities present in the protein sequence/structure. For example,
>  - AAC features are counts of the amino acid types in a sequence
>  - OCPC features are counts of number of 3D contacts a pair of amino acid types have in a structure
>  - Table 1 has such interpretations for other features.
>
> In our one-vs-all classification setup for CATH superfamilies, we have the *following insight*, which is offered by these features *based on the sign of the linear classifier coefficients*. We can infer that the group of feature components corresponding to the positive coefficients of linear SVM are rich in class +1 (a given CATH superfamily). Similarly, the group of feature components corresponding to the negative coefficients are deficient in class +1 (a given CATH superfamily). The features can be further ranked using feature importance scores (such as [MCI, Catav et al., ICML 2021](https://proceedings.mlr.press/v139/catav21a.html); [Tripathi et al., 2020](https://ieeexplore.ieee.org/abstract/document/9378102)). To illustrate this with an example, we have computed the MCI feature importance scores of AAC for superfamily 2.130.10.10 vs 'others' classification. Please consider the figure: [here](https://anonymous.4open.science/r/cath_classification-8A0C/mci_2_130_10_10.png). The *amino acid $S$ with the highest feature importance score is known to be present in a repeating motif found in many of the structures belonging to this superfamily* ([Chaudhari et al., Proteins 2008](https://pubmed.ncbi.nlm.nih.gov/17979191/)).
>
> With the more informative CSIC features, an important feature corresponding to a positive classifier coefficient indicates a type of 3-dimensional contact characteristic of a given superfamily. A CSIC feature vector component, as in Table 1, is interpreted as follows: *'number of contacts an amino acid type $t_i$ has in the structure with another amino acid separated by at least $l$ and at most $u$ residues in the sequence'*. However, efficiently computing feature importance scores for high-dimensional features is challenging. Also, validating the computed feature importance requires wet-lab-based experimentation. These may/may not be done in existing literature for many CATH superfamily structures. Thus, we did not report feature importance scores in the paper. However, we can include some examples in the revision.
>
> Thus, the biological insight can be enabled by the interpretability of the feature components. This is not possible with PLM features.
>
> *We will revise our submission to include the aspects from our above responses which were earlier missing.*

---

> > ### Comment · Reviewer_qdz7 · 2025-11-24
> > **Response to Weaknesses 2**
> >
> > I am fine with the general direction of the response (which also has to be reflected in the manuscript). However, currently the discussion remains at a descriptive level. I think it would help the submission a lot to provide a case study, where you show for 1-2 superfamilies the most discriminative features and to indicate a biological interpretation of them to demonstrate that your method actually provides new insights.
> >
> > To summarize, the most crucial weaknesses from my point of view remain:
> > 1. Case study/biological insights from above
> > 2. Experimental validation for the claim related to AlphaFold structures
> > 3. Scope and MSA-baseline: clearly acknowledging the niche domain, where MSA features are not applicable; and ideally a supplementary task
> > 4. Experimental and statistical analysis: Ideally consider a different classification head that is able to capture feature correlations and indicate statistical significance statements that enable meaningful comparisons between the proposed methods.

---

> ### Comment · Reviewer_qdz7 · 2025-11-24
> **Response to Questions**
>
> 1. I am satisfied with the response. However, the manuscript should also reflect this argument.
> 2. I am not conviced that a linear SVC yields the best compromise between performance and interpretability. Possible alternative could be gradient-boosted decision tree with XAI methods or similar (c.f. also discussion of correlated features below).
> 3. The most critical issue remains the claim of scalability via AlphaFold. A central contribution of your work is the potential for broad application using predicted structures, yet the study is conducted exclusively on experimental structures. This claim requires validation. To be compelling, you need to demonstrate that your contact-based features computed from AlphaFold predictions retain their discriminative power compared to those derived from experimental structures.

---

> ### Comment · Reviewer_qdz7 · 2025-11-24
> **Response to Weaknesses 1**
>
> I appreciate the authors' detailed responses. However, the rebuttal has reinforced several of the original concerns.
>
> * The justification for a single task remains unsatisfactory. The claim that EC classification lacks characteristic features is a hypothesis to be tested, not a premise to avoid the task. Using AlphaFold to apply your method to such a task would be a far stronger demonstration of its scalability and general utility.
>
> * The inapplicability of MSA features for this specific dataset should be explicitly discussed in the manuscript, as it contextualizes your results within a niche where a major class of methods does not apply.
>
> * The issue of feature combination and overfitting may be a limitation of the chosen linear SVM model. Exploring more robust models like Gradient Boosted Trees could handle feature interactions better and provide more reliable feature importance scores.
>
> * While multiple train/test splits were used, this assesses model stability, not the statistical significance of performance differences. I strongly recommend using for example bootstrapping on the test set to assess the statistical significance of performance differences.

---

> ### Author Response · Authors · 2025-12-03
>
> # 1. Case study/biological insight
>
> ## Study-1: CATH superfamily 3.40.640.10 vs 'others' - CSIC-Gaussian features
> Recall, CSIC features are $K \times 20$ dimensional, where the $K$ rows correspond to sequence separation intervals and the 20 columns correspond to amino acid types (please see Table 1 of the main paper). MCI feature importance ([Catav et al., ICML 2021](https://proceedings.mlr.press/v139/catav21a.html)) approximations can be very bad for high-dimensional features. Therefore, we compute row-wise and column-wise feature importance of the $K\times20$ feature matrix.
>  Please see figures here: [[link]](https://anonymous.4open.science/api/repo/cath_classification-8A0C/file/mci_csic_34064010.png?v=44de1c05). The intervals [137, 167] and [169, 205] have the highest row-wise feature importance scores. *These are long-range contacts present in the structures of this superfamily* as highlighted in the contact maps. Please see figure: [[link]](https://anonymous.4open.science/api/repo/cath_classification-8A0C/file/contact_maps_34064010.png?v=bc6853c8). *A study [(Kirsch et al, Protein Science 2002)](https://pmc.ncbi.nlm.nih.gov/articles/PMC2373551/) highlights the role of a long-range contact that falls within this range* [137, 205], in the structure of an aspartate aminotransferase domain that belongs to 3.40.640.10.
>
> ## Study-2: CATH superfamily 2.130.10.10 vs 'others'
>
> ### CSIC-Gaussian
> As in the previous example, we compute row-wise and column-wise MCI feature importance of the $K\times20$ feature matrix.
>  Please see figures here: [[link]](https://anonymous.4open.science/r/cath_classification-8A0C/mci_csic_21301010.png).
>
> (*Row-wise feature importance: intervals*) The interval [4,18] has the highest row-wise feature importance. We consider the contact maps of some domain structures belonging to superfamily 2.130.10.10. Please see the figure here: [[link]](https://anonymous.4open.science/r/cath_classification-8A0C/contact_maps_21301010.png). We observe in the contact maps that the *structures are rich in contacts formed by amino acids with sequence separation in the range* [4,18], and this is *characteristic of [anti-parallel $\beta$-strands](https://en.wikipedia.org/wiki/Protein_contact_map#/media/File:Elements_hb2.jpg) present in [$\beta$-propeller](https://en.wikipedia.org/wiki/Beta-propeller) structures*.
>
> (*Column-wise feature importance: amino acids*) The amino acids $V$ and $T$ have the highest column-wise feature importance (>0.95).  Amino acids $V$ and $T$ are known to be present in a *repeating motif ('$YVTN$') found in many of the structures* belonging to this superfamily ([Chaudhari et al., Proteins 2008](https://pubmed.ncbi.nlm.nih.gov/17979191/)). Overall, the amino acids $Y, V, T$ and $N$ have greater than 0.75 MCI feature scores. Amino acid $S$ with the third highest feature importance score (>0.95) is *present in a known repeating motif ('$SPDG$') found in many of the structures belonging to this superfamily*. Overall, the amino acids $S, P, D$ and $G$ have greater than 0.75 MCI feature scores.
>
> ### AAC features
>
> *(this example was mentioned in our previous response to **'What insight can be gained?'** and is again reiterated here)*
> We have computed the MCI feature importance scores of AAC for superfamily 2.130.10.10 vs 'others' classification. Please see [figure [link]](https://anonymous.4open.science/r/cath_classification-8A0C/mci_2_130_10_10.png). The amino acid $S, T$ and $D$ with the highest feature importance scores (>0.75) are *present in a known repeating motifs ('$SPDG$' and '$YVTN$') found in many of the structures belonging to this superfamily* ([Chaudhari et al., Proteins 2008](https://pubmed.ncbi.nlm.nih.gov/17979191/)).
>
> Thus, *the CSIC features*, which utilise contact sequence separation information, *offer a more nuanced interpretability compared to other hand-crafted features or PLM-based features*. Structure-level interpretations of characteristic features in a superfamily are possible using the CSIC features. These can be further tested through directed wet-lab experiments to gain biological insights, such as the stability of the structure when these characteristic features are manipulated. However, such wet-lab experiments to validate feature importance are beyond the scope of the current study.
>
> *(1/3)(reponses continued in next comment)*

---

> ### Author Response · Authors · 2025-12-03
>
> # 2. Experimental validation with AlphaFold structures for CSIC features
>
> We wish to point out that an independent verification has found the overall performance of AlphaFold structure prediction to be extremely high; verification is done against wet-lab experimental data as a part of [CASP](https://predictioncenter.org/). CASP (Critical Assessment of Protein Structure Prediction) is a community-wide, international experiment for protein structure prediction that takes place every two years since 1994. '*Prediction methods are assessed on the basis of the analysis of a large number of blind predictions of protein structures*'[[Reference](https://predictioncenter.org/)].  AlphaFold was awarded the 2024 Chemistry Nobel Prize.
>
> In addition, we created a dataset of AlphaFold predicted domain structures from the [TED database (link)](https://ted.cathdb.info/). We collected 5350 structures in total. These proteins do not have experimentally determined structures available. The domain identifiers for these structures can be viewed here: [[link](https://anonymous.4open.science/r/cath_classification-8A0C/ted_alphafold_rep50.csv)]. These included 50 structures for each of the 45 superfamilies (total $45\times50=2250$  structures). And 3100 structures did not belong to any of the 45 superfamilies. We tested each of the binary one-vs-all classifiers trained using CSIC features from our original dataset on the new curated AlphaFold structure dataset. The average classification AM score across the 45 superfamilies is $85.2 \pm 6.46$ (mean $\pm$ standard-deviation). Thus, we *do not observe a significant drop in classification performance when the CSIC feature is computed from AlphaFold structures.*
>
>
>
> # 3. Supplementary task: EC classification
>
> > '*'EC classification lacks characteristic features' is a hypothesis to be tested, not a premise to avoid the task*'
>
> We respectfully disagree with the reviewer's above comment. It is a well-established fact in literature that enzymes with the same EC number do not necessarily share characteristic sequence or structural features. Please see references: ([Omelchenko et al., Biol Direct. 2010](https://pmc.ncbi.nlm.nih.gov/articles/PMC2876114/); [Riziotis et al., FEBS J. 2024](https://pmc.ncbi.nlm.nih.gov/articles/PMC11796326/)).
>
> *(2/3)(reponses continued in next comment)*

---

> ### Author Response · Authors · 2025-12-03
>
> # 4. Experimental and statistical analysis
>
> ## Using a different classification head - Gradient boosted trees
>
> The class of decision trees can have arbitrarily nonlinear decision boundaries and can be considered to be a richer hypothesis class in comparison to linear classifiers [(Leboeuf et al, Neurips 2020)](https://proceedings.neurips.cc/paper/2020/hash/d2a10b0bd670e442b1d3caa3fbf9e695-Abstract.html). Thus, *we do not expect gradient boosted trees to have lesser overfitting*. In fact, *we verify this assumption using computations discussed below*.
>
> We train histogram-based gradient boosting classification trees (HGBCT), which is recommended for large datasets. We use the scikit-learn library [[link]](https://scikit-learn.org/stable/modules/generated/sklearn.ensemble.HistGradientBoostingClassifier.html). We use gridsearch to tune the hyperparameters from the ranges defined below,
>
> tuning_params = {
>     'learning_rate': [0.01, 0.1, 0.2],
>     'max_iter': [100, 200, 300],
>     'max_depth': [3, 5, 7],
>     'l2_regularization': [0.0, 0.1, 1.0],
>     'min_samples_leaf': [10, 20, 40],
>     'max_bins': [64, 128, 255]
>   }
>
> The HGBCT classification performances are shown in the table below. The average train/val/test AM scores over 5 random splits are again averaged across the 45 superfamilies. Similarly, the standard deviations (s.d.) of train/val/test AM scores over 5 random splits are again averaged across the 45 superfamilies.
>
> | AM | Metric | AAC | DPC | 2OAAC | OCPC | CSIC-Gauss | CSIC-Gamm | PB-Attn | PB-Emb |
> | :--- | :--- | :--- | :--- | :--- | :--- | :--- | :--- | :--- | :--- |
> | **Train** | Avg. | 91.9 | 91.7 | 93.2 | 93.6 | 96.5 | 96.6 | 94.2 | 97.8 |
> | | (s.d.) | (3.32) | (3.52) | (2.67) | (2.35) | (2.12) | (2.05) | (2.67) | (1.21) |
> | **Val** | Avg. | 84.9 | 79.2 | 85.9 | 86.0 | 91.7 | 92.1 | 88.1 | 91.3 |
> | | (s.d.) | (5.22) | (6.73) | (4.87) | (5.22) | (3.77) | (4.45) | (4.93) | (4.34) |
> | **Test** | Avg. | 79.9 | 73.9 | 80.3 | 80.2 | 87.3 | 87.6 | 81.2 | 86.9 |
> | | (s.d.) | (5.79) | (7.52) | (6.23) | (5.49) | (4.26) | (5.08) | (6.05) | (4.71) |
>
>
>
> Observations based on the above:
>  - We find the test classification performance of HGBCT using hand-crafted features is similar to that of linear SVM (please see Table 2 of the main paper).
>
>  - For ProtBERT-based features, we observe some overfitting. The overfitting is more significant for the ProtBERT-Emb features (please see in comparison to Table 2 of the main paper). The test score drops from 96.5 using a linear SVM to 86.9 with HGBCT for ProtBERT-Emb features.
>
>  - For all features except CSIC-Gauss, the standard deviation of the HGBCT test scores is greater than that of linear SVM scores (please see Table 2 of the main paper).
>
> Thus, *the gradient-boosted trees classifier does not help us overcome overfitting.*
>
> ## Statistical significance of performance differences using bootstrapping
>
> We have now performed bootstrapping on the test set to compute a 95\% confidence interval for the test AM score differences of the linear SVMs trained using different feature types. We used a bootstrap sample size of 1000. Please see figure [[link]](https://anonymous.4open.science/r/cath_classification-8A0C/CSIC_bootstrap_diffs.png). Based on these confidence intervals, we have ranked the features for each superfamily. Please see this in the table: [[link]](https://anonymous.4open.science/api/repo/cath_classification-8A0C/file/feature_comparison_table_bootstrap.pdf?v=f4f340bc).
>
> We observe that the ProtBERT-Emb feature consistently outperforms other features across the 45 superfamilies. ProtBERT-Attn and CSIC features perform comparably across the 45 superfamilies, while the rest of the hand-crafted features have relatively lower classification performances compared to them.
>
> Thus, *these experiments concur with our conclusion that CSIC features strike a balance between performance and interpretability*.
>
> *(3/3)*

---

### Official Review · Reviewer_4jFd · 2025-10-24

**Soundness:** 3
**Presentation:** 4
**Contribution:** 3
**Rating:** 6
**Confidence:** 3

**Summary:**

This paper touches a tradeoff between the interpretability and performance, and their findings were insightful especially to the domains where interpretability is important. They trained LinearSVC (in some cases SGDClassifier) with features either by hand-crafted engineering or PLMs, and compared the results in predicting 45 CATH superfamilies. The authors provided insight into how an interpretable features would achieve a comparable performances in the down streaming tasks.

**Strengths:**

1. The comparison processes were rigorous and the authors used both sequence and structure based hand-crafted features.
2. The authors provided novel structure features, that could be useful to the biocomputation community, also the ProtBERT-Attn could also be useful even if it is not interpretable.
3. The authors covered the imbalance problems. The dataset is diverse, covering 45 CATH superfamilies.
4. The authors ran five random splits which made the experiment more rigiorous.
5. Easy to follow and understand the methodology.

**Weaknesses:**

1. The scope of this paper is my biggest concern. The paper aims to achieve a balance between performance and interpretability of PLMs. However, only the ProtBERT was evaluated. The scope of PML comparison is somehow limited, making me worried about whether their observations/results could be generalized when using other PLMs such as the ESM family or ProtT5. By doing so they can cover more training set of LMs and more dimensions.
2. For TPC and 30AAC, they were using SGDClassifier, how would that compared with the cases when LMs also give high-dimensional features? (Related to Weaknesses #1)
3. In most hand-crafted feature scenarios, the model is overfitting.
4. Complexity to calculate CSIC.
5. Missing some ablation studies: for example, the authors' conclusions would be more solid if they can ablate the ProtBERT-Attn Feature (16 is the total), or ablating the CSIC itself (for instance, Intervals K)
6. (Minor) Table 5 doesn't have the dimensions for the features. Although table 1 has such information, table 5 would be more readable if the authors could indicate which features are high-dimensional or not. In addition, the order the features in Table 5 could be consistent with Table 1.

**Questions:**

1. Can you combine multiple features? How will that affect the performances of the LinearSVC/ SGDClassifier.
2. How might your experiments be generalized to non-bio-computation domains?

---

> ### Author Response · Authors · 2025-11-21
>
> *We thank the reviewer for taking time to review our submission. Please find below our answers to the questions asked*
>
> # Questions
>
> ## 1. Combining multiple features
> We observe that with the hand-crafted features DPC, 2OAAC, TPC and 3OAAC, there is significant model overfitting. We observed increased overfitting when combining these features. When combining AAC and CSIC features, the hand-crafted features with the least overfitting, we do not see a significant difference in test scores.
>
> ## 2. Generalisation to non-bio-computation domains
>
> In this work, we focus on the more traditional art of hand-crafted feature engineering, which is interpretable and tailored to the specific task of CATH superfamily classification. In general, for non-bio-computation domains, one may wish to engineer interpretable features for a given task to obtain insights beyond black-box predictive power. This would require incorporating the necessary domain knowledge. For example, *we felt the distribution of sequence separation lengths of 3-dimensional residue contacts to be a characteristic feature of protein structures and hence utilised this information to engineer the CSIC features*. This was substantiated by the classification performance of these features, which is comparable to PLM-based features.
>
> # Weaknesses
>
> ## 1. Only ProtBERT evaluated - PLM comparison limited
>
> We use ProtBERT as a representative PLM to illustrate the high classification performance that PLMs achieve on this task. ProtBERT achieves an average test score of 96.5\% across the 45 superfamilies. This is already a high score, and using any other PLM can achieve rather minor - 4.5\% - improvement on this. Thus, we believe that ProtBERT sufficiently demonstrates the high predictive power of PLMs. Furthermore, the notion of interpretability is largely similar across popular PLMs, such as ESM and ProtT5. This is by finding correlations between attention values and known protein properties ([Simon \& Zou, Nature Methods 2025](https://www.nature.com/articles/s41592-025-02836-7), [Vig et al., ICLR 2021](https://openreview.net/forum?id=YWtLZvLmud7)).
>
> Thus, we believe that ProtBERT adequately represents the PLMs on both performance and interpretability objectives. Hence, other PLMs were not used in this study as the focus of the work is towards engineering interpretable features tailored for this task, which can achieve competitive performance.
>
> ## 2. Using SGD classifier for TPC and 3OAAC, and higher-dimensional LM features
>
> *(We are unsure if we understood this weakness correctly; however, below is our response based on our understanding. Please let us know for further clarification.)*
>
> Since the 3OAAC and TPC features are 8000-dimensional, and with the number of training samples exceeding 23,000, the linear SVM quadratic program took longer to solve due to the large number of variables. Thus, we used an SGD classifier with hinge loss and a mini-batch size of 10,000 to find a faster approximate solution. Since the ProtBERT features are 1024-dimensional ($\sim$ 8 times less), the SVM quadratic program was solved relatively quickly.
>
> In summary, for all the features, the classifier trained is a linear SVM that is obtained by minimising the regularised hinge loss objective. The choice of optimization technique was based on the size of the dataset - SGD or liblinear via scikit-learn.
>
>
>
> *(Responses to other weaknesses are in the next comment)*

---

> ### Author Response · Authors · 2025-11-21
>
> # Weaknesses
>
> ## 3. Overfitting in hand-crafted features
> Yes, we also observed overfitting, i.e. significant gap in the train and test scores, for all hand-crafted features except AAC. This gap is higher for DPC, 2OAAC, TPC, 3OAAC and OCPC features. Our understanding is this: This could be due to the higher feature dimensions of these feature (400 to 8000) in comparison to the lower sample size of the minority samples (100 to 873). The insignificant train/test score gap for AAC could be due to its low dimensions (20). For CSIC features, the feature dimensions are data-dependent and range from 80 to 260. The train/test gap for CSIC is relatively lower compared to the high-dimensional ($>400$) hand-crafted features.
>
>  Another aspect here is that, any pair of sequences in our dataset have less than 35\% sequence identity (Please see Section 3.1 *Dataset diversity*). As a result, the *test sequences have less than 35\% identity with the training sequences*. This makes *generalization of classification performance on test set challenging*.
>
>
> Interestingly, we find the significantly less overfitting in 1024-dimensional ProtBERT-Emb features. This could be as a consequence of the self-supervised pre-training on a large number of sequences. This pre-training could have enabled a good approximation of the naturally occurring sequence data manifold. The test sequences of our experiments could have also been part of the pre-training dataset (the [BFD](https://bfd.mmseqs.com/) dataset).
>
> ## 4. Complexity to calculate CSIC
> We thank the Reviewer for this good suggestion; we first summarise CSIC   calculations we reported in the paper and  now report the computational complexity.
>
>
> We discuss in Section 2.2.2 that for CSIC feature computation, a user-defined input, i.e. a set of sequence separation intervals $\mathcal{I}$ is required. In Section 4.1.2, we define the set $\mathcal{I}$ in a data-driven manner using Gaussian mixtures to get CSIC-Gaussian features.
> Thus, the computationally expensive step for CSIC-Gaussian feature calculation is the expectation-maximisation (EM) algorithm for Gaussian mixture modelling (GMM). Since the GMM is on a 1-dimensional distribution (Please see Figure 3 in the main paper), one EM step has a complexity of $\mathcal{O}(n\cdot K)$ ([Pinto \& Engel, 2015](https://pmc.ncbi.nlm.nih.gov/articles/PMC4596621/)), where $n$ is the number of samples and $K$ is the number of components for GMM. Note that the samples $n$ for the GMM are the total number of contacts from all the structures of a given CATH superfamily. This is because the GMM is fitted to the distribution of contact sequence length separations. The mean $\pm$ standard deviation of $n$ is $525 \pm 341$. We determine the number of Gaussian components, $K$, using a grid search with $K$ taking values from 2 to 14. The best $K$ is determined based on the [Akaike information criterion (AIC)](https://en.wikipedia.org/wiki/Akaike_information_criterion). We run the EM algorithm for at most $10^4$ steps or until the average gain of the log likelihood lower bound is below $10^{-4}$.
>
> ## 5. Missing ablation studies
>
> We thank the Reviewer for this good suggestion on ablation studies, which we have implemented.
>
> For the CSIC-Gaussian feature computation, the hypermeter $K$ was tuned earlier by us, but we forgot to mention in the main paper. Here are the details: we determine the number of Gaussian components, $K$, using a grid search with $K$ taking values from 2 to 14. The best $K$ is determined based on the [Akaike information criterion (AIC)](https://en.wikipedia.org/wiki/Akaike_information_criterion).
>
> We have now conducted an ablation study for various values of $K$ using CSIC-Gaussian features. The following figure: [link](https://anonymous.4open.science/r/cath_classification-8A0C/CSIC_K_ablation.png), shows the train/test AM scores for different values of $K$ and the $K$ value that was chosen based on gridsearch for 6 CATH superfamilies.
>
> We do not conduct an ablation study on the 16 attention heads of the ProtBERT-Attn feature, as suggested by the reviewer, because the number of attention heads is fixed by the pre-trained model. One may choose only a subset of attention heads, but there are $2^{16}$ subsets to choose from.
>
> *We will revise our submission to include the aspects from our above responses which were earlier missing.*

---

### Official Review · Reviewer_taws · 2025-10-30

**Soundness:** 1
**Presentation:** 2
**Contribution:** 2
**Rating:** 2
**Confidence:** 2

**Summary:**

This paper discusses the trade-off between manual features and protein language model (PLM) features in protein superfamily classification tasks in terms of prediction performance and interpretability, and proposes a new feature engineering method to balance the two.

**Strengths:**

1. By employing various feature engineering techniques, this study analyzed the encoding capacity of protein sequences for genetic information.
2.proposed one-vs-all classifiers to predict the CATH homologous superfamily of a protein domain.

**Weaknesses:**

1.The article's structure and mathematical notation are somewhat disorganized, making it difficult for readers to grasp the core methodology.
2.The study lacks sufficient experiments, fails to analyze downstream applications, and does not conduct ablation experiments on the method itself.

**Questions:**

How can the proposed study be integrated with existing LLM-based AI technologies for protein understanding and generation?

---

> ### Author Response · Authors · 2025-11-21
>
> *We thank the reviewer for taking time to review our submission. Please find below our answers to the questions asked.*
>
> # Questions
>
> ## Integration with LLM technologies for protein understanding and generation
>
> In this work, we engineer interpretable hand-crafted features tailored for the task of CATH superfamily classification. Features that are characteristic of a given superfamily can be inferred using feature importance scores ([MCI, Catav et al., ICML 2021](https://proceedings.mlr.press/v139/catav21a.html); [Tripathi et al., 2020](https://ieeexplore.ieee.org/abstract/document/9378102)). More details illustrated with an example superfamily are provided in the response below to *'Downstream applications'*. This enables *protein understanding*. Additionally, knowing the *characteristic features of a given superfamily can be useful in protein design* for creating proteins of a specific fold type as defined by CATH IDs.
>
> Currently, we are unsure how LLM integration can be achieved, and it appears to be beyond the scope of the current study.
>
> *(clarifications to weaknesses are continued in the next comment)*

---

> ### Author Response · Authors · 2025-11-21
>
> # Weaknesses
>
> We thank the Reviewer taws for pointing out the role of ablation studies.
>
> ## 1. Article structure and notation disorganised
>
> It is not clear to us in what way the structure and notation were not properly organised. Please suggest specific portions  (that would be helpful to us), and we can surely incorporate them.
>
> ## 2. Lacks sufficient experiments
>
> We have to respectfully disagree with the reviewer.  We have considered 45 CATH super-families and 8 hand-crafted features for each of them, in addition to 2 feature sets engineered on protein language models (PLMs). We believe that these are adequate as a set of comprehensive computational experiments, involving numerous datasets and feature sets, making our study thorough and robust. We would also point out that these *datasets have varied sizes, class imbalances, and biological diversity (Section 3.1)*. Again, if specific suggestions are provided, we can consider executing them.
>
> ### Downstream application
>
> We point out in line 77 of the main paper that a useful downstream task is to '*identify features that are characteristic of a CATH superfamily using feature importance measures*'.
>
> **How is it useful?** In our one-vs-all classification setup for CATH superfamilies, we can infer that the group of feature components corresponding to the positive coefficients are rich in class +1 (a given CATH superfamily). Similarly, the group of feature components corresponding to the negative coefficients are deficient in class +1 (a given CATH superfamily). The features can be further ranked using feature importance scores, such as marginal contribution feature importance ([MCI, Catav et al., ICML 2021](https://proceedings.mlr.press/v139/catav21a.html); [Tripathi et al., 2020](https://ieeexplore.ieee.org/abstract/document/9378102)). To illustrate this with an example, we have computed the MCI feature importance scores of AAC for superfamily 2.130.10.10 vs 'others' classification. See [figure](https://anonymous.4open.science/r/cath_classification-8A0C/mci_2_130_10_10.png). The amino acid $S$ with the highest feature importance score is known to be present in a repeating motif found in many of the structures belonging to this superfamily ([Chaudhari et al., Proteins 2008](https://pubmed.ncbi.nlm.nih.gov/17979191/)).
>
>
> With the more informative CSIC features, an important feature corresponding to a positive classifier coefficient indicates a type of 3-dimensional contact characteristic of a given superfamily. A CSIC feature vector component, as in Table 1, is interpreted as follows: *'number of contacts an amino acid type $t_i$ has in the structure with another amino acid separated by at least $l$ and at most $u$ residues in the sequence'*. However, efficiently computing feature importance scores for high-dimensional features is challenging. Also, validating the computed feature importance requires wet-lab-based experimentation. These may/may not be available in existing literature for many CATH superfamily structures. Thus, we did not report feature importance scores in the paper. However, we can include some examples in the revision.
>
> Thus, biological insights can be gained by leveraging the interpretability of the feature components. This is not possible with PLM features.
>
> ### Ablation study
>
> For the CSIC-Gaussian feature computation, the hypermeter $K$ was tuned earlier by us, but we forgot to mention in the main paper. Here are the details: we determine the number of Gaussian components, $K$, using a grid search with $K$ taking values from 2 to 14. The best $K$ is determined based on the [Akaike information criterion (AIC)](https://en.wikipedia.org/wiki/Akaike_information_criterion).
>
> We have now conducted an ablation study for various values of $K$ using CSIC-Gaussian features. The following figure: [link](https://anonymous.4open.science/r/cath_classification-8A0C/CSIC_K_ablation.png), shows the train/test AM scores for different values of $K$ and the $K$ value that was chosen based on gridsearch for 6 CATH superfamilies.
>
> *We will revise our submission to include the aspects from our above responses which were earlier missing.*

---

> ### Comment · Reviewer_taws · 2025-11-28
>
> I appreciate the author's response to my question, the explanation of the weakness, and the claim regarding optimization. I believe these have partially addressed my initial doubts when reading the article.
> I still believe that the current article lacks sufficient experimental analysis, and fails to provide a comprehensive evaluation of hyperparameters and feature engineering approaches from multiple perspectives.
> Some formulaic expressions, such as x^{kPC}, x^{kOAAC}, and x^{ProtBERT-Attn}, lack elegance in their notation.
> The reference format is inconsistent. Check carefully.
> Regarding its application scope, I endorse the use of protein superfamily classification, yet its biological significance remains insufficiently explicit.
> In conclusion, I am willing to maintain my current rating.

---

> > ### Author Response · Authors · 2025-12-03
> >
> > ## Evaluation of hyperparameters
> > We have only two hyperparameters to tune in our method. These are discussed below.
> >
> > **SVM regularization parameter $C$**. As stated in line 379 of the main paper, we use a validation set to tune $C$. $C$ is chose using gridsearch from $\{0.01, 0.1, 1, 10, 100\}$. If $C=0.01$ is chosen then we again check if $C=0.001$ gives performance improvement on validation set. Similarly, if $C=100$ is chosen we check at $C=1000$.
> >
> > **Number of intervals $K$ for CSIC**. As stated in our earlier response '*Ablation study*', we use Akaike information criteria to find the best $K$ from the values 2 to 14.
> >
> > These are the only two hyperparameters in our study. We will include these details in the revised manuscript.
> >
> > ## Evaluation of feature engineering approaches from multiple perspectives
> >
> > In this work, we aimed to evaluate features from two perspectives: predictive performance and interpretability. Since there is no metric for interpretability, we discuss interpretability aspects of the features in Table 1 of the main paper. The predictive performance of the features on the CATH superfamily classification task is discussed in Section 5 (Results and Discussion) of the main paper.
> >
> > ## Biological significance remains insufficiently explicit
> >
> > We have now done a case study on the biological significance of the features for two superfamilies. Please see our response to reviewer *qdz7* with the title - '***1. Case study/biological insight***'.
> >
> > We illustrate through these case studies that the CSIC features *offer a more nuanced interpretability compared to other hand-crafted features or PLM-based features*. Structure-level interpretations of characteristic features in a superfamily are possible using the CSIC features. These can be further tested through directed wet-lab experiments to gain biological insights, such as the stability of the structure when these characteristic features are manipulated. However, such wet-lab experiments to validate feature importance are beyond the scope of the current study.
> >
> > ## Regarding references and notations
> >
> > We have fixed the references, which contained: one duplicate entry and author names not expanded for one other reference.
> >
> > Regarding the notation. Since other reviewers have found it (reviewer xiRu) '***well-structured and easy to follow***'  and (reviewer 4jFd) '***easy to follow and understand the methodology***' , we would like to stick to our current notations. Any suggestions on how to improve them are welcome.

---

### Official Review · Reviewer_xiRu · 2025-10-31

**Soundness:** 2
**Presentation:** 2
**Contribution:** 1
**Rating:** 2
**Confidence:** 5

**Summary:**

This paper investigates the performance versus interpretability trade-off in protein superfamily classification by comparing nine hand-crafted features versus protein language model features (ProtBERT). The authors conclude that while PLMs offer high predictive accuracy, carefully engineered interpretable features like CSIC can balance performance and interpretability.

**Strengths:**

1. The paper is well-structured and easy to follow.
2. I appreciate the effort in designing the evaluations for class-imbalance protein predictions, which is well-motivated in real life.

**Weaknesses:**

Major Issues

The experimental setup is far too narrow to support the paper's broad claims. The authors argue there's a fundamental trade-off between performance and interpretability in protein modeling, but they only test this on a single task with a single PLM:

* CATH superfamily classification is a simplified sequence similarity problem. The conclusion cannot be generalized to other tasks like  binding affinity and structure predictions. The paper can't properly claim a "fundamental trade-off" exists across computational biology when it  only looked at one relatively simple classification problem.

* only ProtBERT:  ESM is SOTA on most benchmarks and is what people actually use in practice. Testing against an older, weaker model doesn't tell us much about the real performance gap.

This experiment just shows ProtBERT beats some hand-crafted features on superfamily classification. The conclusions about performance vs. interpretability being a universal trade-off cannot be drawn given this limited evidence.

In addition, the paper claim hand-crafted features are interpretable, but never demonstrate this. Where's the biological insight? What features distinguish different superfamilies? Without showing this, what is the interpretability?

Overall, the execution of the paper lacks the experimental rigor and technical depth expected for a venue like iclr.

**Questions:**

See weakness.

---

> ### Author Response · Authors · 2025-11-21
>
> *We thank the reviewer for taking the time to review our submission. Please find below our answers to the questions asked.*
>
> ## On the CATH superfamily classification task
>
> > '*CATH superfamily classification is a simplified sequence similarity problem.*'
>
> We respectfully disagree with this comment. CATH superfamilies are not totally assigned by sequence similarity alone. It heavily *relies on manual curation and structural information* and requires validation (See [Orengo et al., 1997](https://pubmed.ncbi.nlm.nih.gov/9309224/)). As discussed in Section 3.1 (Dataset diversity), for the dataset considered in our experiments, the sequences are filtered such that *'No two sequences have more than 35\% sequence identity'*. The following figures, [inside-fig](https://anonymous.4open.science/r/cath_classification-8A0C/cath_inside.png) \& [outside-fig](https://anonymous.4open.science/r/cath_classification-8A0C/cath_outside.png),  from the [CATH database](https://www.cathdb.info/version/latest/superfamily/1.10.10.10/structure) illustrate the within-family and outside-family sequence identity for CATH ID 1.10.10.10. The *within-family* sequence identities are largely between 0-40\% with most of the concentration being at 0-20\%. The *outside family* sequence identities are largely between 0-20\%. This further emphasises that it is not a simple sequence similarity problem. Furthermore, due to the 35\% sequence identity cutoff, *the test sequences have less than 35\% sequence identity with the training sequences*, making generalisation of classification performance on test data difficult. This is reflected in the train/test score differences in Table 3 of the main paper. Thus, this is a hard classification task.
>
> In this context, we would also like to mention that this dataset is very rich in its information/statistical content.  So, for the classification task that we considered, we had to use  existing and novel feature engineering methods to encode varying levels of information - from composition to sequence-order  to structural information (*novel features - OCPC \& CSIC*).
>
>
> *(responses continued in next comment)*

---

> ### Author Response · Authors · 2025-11-21
>
> ## Performance vs interpretability shown only for a single task
>
> Below, we highlight that the performance vs. interpretability tradeoff in PLMs is supported by existing literature. We also discuss the contribution of our submission in this context.
>
> **Predictive performance of PLMs:** The high predictive performance of PLMs on various tasks and their prevalence in biological research are *well-documented in the existing literature*. Some references are below:
>
>  - ([Weissenow \& Rost, 2025](https://www.sciencedirect.com/science/article/pii/S0959440X25000156)), page 3, subsection *Embeddings competitive in accuracy*. Highlights the high predictive performance of PLMs.
>  - ([Pokharel et al., 2025](https://link.springer.com/protocol/10.1007/978-1-0716-4623-6_1)). Highlights the prevalence of PLMs in comp bio.
>  - ([Simon et al., 2024](https://www.nature.com/articles/s41592-024-02354-y)). Highlights the prevalence of PLMs in comp bio.
>
> (Pokharel et al., 2025; Weissenow \& Rost, 2025) have *already been cited in lines 42 and 46* of page 1 in the main paper.
>
> **Interpretability of PLMs:** The interpretation or explanation of PLMs or language models in general is *done by examining the attention matrices*. After pretraining the models, many works *highlight correlations between attention values and known protein properties* ([Simon \& Zou, Nature Methods 2025](https://www.nature.com/articles/s41592-025-02836-7), [Vig et al., ICLR 2021](https://openreview.net/forum?id=YWtLZvLmud7)). However, this is an emergent phenomenon from unsupervised learning of the data manifold of available sequences. There is *no established causal relationship between the attention values and the respective property* (Please see *'Limitations'* in [Simon \& Zou, 2025](https://www.nature.com/articles/s41592-025-02836-7)).
>
> Recently, there has been a *growing debate over the use of attention for interpretability*. See references below,
>
>  - [(Jain \& Wallace, NAACL 2019) Attention is not Explanation](https://aclanthology.org/N19-1357/)
>  - [(Pruthi et al., ACL 2020) Learning to Deceive with Attention-Based Explanations](https://aclanthology.org/2020.acl-main.432/)
>  - [(Hassid et al., EMNLP 2022) How Much Does Attention Actually Attend? Questioning the Importance of Attention in Pretrained Transformers](https://aclanthology.org/2022.findings-emnlp.101/)
>  - [(Bibal et al., ACL 2022) Is Attention Explanation? An Introduction to the Debate](https://aclanthology.org/2022.acl-long.269/)
>
> In summary, the embeddings of PLMs or the attention values *do not have direct domain knowledge-based interpretability by design*.
>
>
> **Contribution of this work:** In this work, we focus on the task of CATH superfamily classification and engineer 8 interpretable features tailored for this task. This is in contrast to PLM features, which, although lacking interpretability, achieve high accuracy on varied tasks. The *proposed hand-crafted CSIC features have direct feature interpretability* (Please see Table 1 of the main paper) and achieve good performance on the given task. This performance is achieved using a simple *linear classifier* with a suitable loss function. In contrast, PLM-based features are a highly non-linear (uninterpretable) transformation of the domain-interpretable raw sequence data. Some additional details are presented in the response below to: '*What insights can be gained?*'.
>
>
>
>
> ## Use of single PLM - ProtBERT
> Here, ProtBERT is used as a *representative PLM to illustrate the high classification performance* that can be achieved on this task. As shown in Table 2, using ProtBERT, an average test score of 96.5\% is obtained. This is already a high score, and using any other PLM can achieve rather minor - 4.5\% - improvement on this. We believe the performance of *ProtBERT sufficiently demonstrates the high predictive power of PLMs*, while the notion of *interpretability is largely similar across popular PLMs*. That is by finding correlations between attention values and known protein properties ([Simon \& Zou, Nature Methods 2025](https://www.nature.com/articles/s41592-025-02836-7), [Vig et al., ICLR 2021](https://openreview.net/forum?id=YWtLZvLmud7)). So, other PLMs were not used in this study.
>
>
>
>
> *(responses continued in next comment)*

---

> ### Author Response · Authors · 2025-11-21
>
> ## Interpretability and insight using hand-crafted features
>
> As already discussed in Table 1 of the main paper, *each component of the hand-crafted feature vector has a domain-knowledge-based interpretation.*
>
> **What insight can be gained?**
>
>
> All the hand-crafted features are positive counts of interpretable quantities present in the protein sequence/structure. For example:
>  - AAC features are counts of the amino acid types in a sequence.
>  - OCPC features are counts of number of 3D contacts a pair of amino acid types have in a structure.
>  - Table 1 has such interpretations for other features.
>
> In our one-vs-all classification setup for CATH superfamilies, we gain the following insight, which is derived from these features based on the sign of the linear classifier coefficients. We can *infer that the group of feature components corresponding to the positive coefficients are rich in class +1* (a given CATH superfamily). Similarly, the group of feature components corresponding to the negative coefficients are deficient in class +1 (a given CATH superfamily). The *features can be further ranked* using feature importance scores, such as marginal contribution feature importance ([MCI, Catav et al., ICML 2021](https://proceedings.mlr.press/v139/catav21a.html); [Tripathi et al., 2020](https://ieeexplore.ieee.org/abstract/document/9378102)). To illustrate this with an example, we have computed the MCI feature importance scores of AAC for superfamily 2.130.10.10 vs 'others' classification. See [figure](https://anonymous.4open.science/r/cath_classification-8A0C/mci_2_130_10_10.png). The *amino acid $S$ with the highest feature importance score is known to be present in a repeating motif found in many of the structures belonging to this superfamily* ([Chaudhari et al., Proteins 2008](https://pubmed.ncbi.nlm.nih.gov/17979191/)).
>
>
> With the more informative CSIC features, an important feature corresponding to a positive classifier coefficient indicates a type of 3-dimensional contact characteristic of a given superfamily. A CSIC feature vector component, as in Table 1, is interpreted as follows: *'number of contacts an amino acid type $t_i$ has in the structure with another amino acid separated by at least $l$ and at most $u$ residues in the sequence'*. However, efficiently *computing feature importance scores for high-dimensional features is challenging*. Also, validating the computed feature importance requires wet-lab-based experimentation. These may/may not be available in existing literature for many CATH superfamily structures. Thus, we did not report feature importance scores in the paper. However, we can include some examples in the revision.
>
> Thus, biological insights can be gained by leveraging the interpretability of the feature components. This is not possible with PLM features.
>
>
> *We will revise our submission to include the aspects from our above responses which were earlier missing.*

---

### Author Response · Authors · 2025-12-03
**Summary of key responses**

We thank all the reviewers for their questions, suggestions, and interest in this work. We especially thank the reviewers who have highlighted the paper's strengths while also pointing out its weaknesses in a constructive manner.

We believe that we have adequately responded to each of their questions.  While we have attempted to thoroughly address each of the points raised by them, we would like to highlight some of the major points that will be reflected in our revised paper.

 - **[Single Task and its difficulty]** - *(Reviewers xiRu and qdz7)*: We will include details on why the CATH superfamily classification is a difficult task and why we didn't consider other tasks like EC number and GO label prediction.

 - **[Use of single PLM - ProtBERT]** - *(Reviewers xiRu, 4jFd and qdz7)*: We will highlight that ProtBERT sufficiently demonstrates the high predictive power of PLMs. Also, we will discuss the lack of interpretability across PLMs in general.

 - **[Interpretability  / Biological insights using hand-crafted features]** - *(Reviewers xiRu, taws and qdz7)*: We will include the case studies that we have now conducted on two superfamilies.

 - **[Feature Combinations]** - *(Reviewers 4jFd and qdz7)*: We will include our observation that feature combinations lead to increased overfitting.

 - **[Ablation study]** - *(Reviewers taws and 4jFd)*: We will include the ablation study that we have now conducted for the CSIC hyperparameter $K$ (number of intervals).

 - **[Complexity to calculate CSIC]** - *(Reviewer 4jFd)*: we will include the details regarding CSIC computation complexity.

 - **[Scalability with AlphaFold for CSIC]** - *(Reviewer qdz7)*: We will include details of our new experiments to validate CSIC feature performance on AlphaFold predicted structures.

 - **[Statistical significance of performance differences]** - *(Reviewer qdz7)*: We will include the experiments that we have now conducted using bootstrapping to find statistical significance of performance differences.

 - *Other editorial comments will be taken care of.*

As suggested by reviewer qdz7, we will move some details of the feature definitions to the appendix, which will provide sufficient space to accommodate necessary additional experiments in the main text (within 10 pages).  We will make our revised paper with the above changes available at this anonymous [link](https://anonymous.4open.science/api/repo/cath_classification-8A0C/file/iclr_revision.pdf?v=003bdce9). The revision will be finalised within a week (Dec 10).

We once again thank the reviewers. The discussion and suggested additional experiments have helped to clarify the goal of the paper, while also validating its conclusions and reemphasizing its contributions.

---

### Meta-Review · Area_Chair_DkUZ · 2026-01-06

**Summary:**

All reviewers recommend rejection or raise major concerns. The experiments are limited to a single task with a single PLM (xiRu, qdz7, 4jFd). The author rebuttals do cite more supporting references though, seemingly did not fully address the concern. In addition, the reviewer taws has raised concerns on the structure and mathematical notations, not sufficient downstream applications and ablation studies. A hyperparameter analysis on K seems not enough, the ablation study should be performed to validate the effects of all major novel ideas and components.

**Reviewer Concerns:**

The same as above.

**Reviewer Scores:**

The four reviewers gave 6, 2, 2, 2 respectively, the author rebuttals did not fully address the concerns.

---

### Decision · Program_Chairs · 2026-01-26

Reject